# Mode Collapse in Variational Deep Gaussian Processes

**Francisco Javier Sáez-Maldonado**
Universidad de Granada
`fjaviersaezm@ugr.es`

**Juan Maroñas**
CUNEF Universidad
Universidad Autónoma de Madrid
`juan.maronas@cunef.edu`

**Daniel Hernández-Lobato**
Universidad Autónoma de Madrid
`daniel.hernandez@uam.es`

## 1 Introduction

Gaussian Processes (GPs)[1] are flexible non-parametric models that have shown promising results across many applications, such as molecule optimization [2, 3] or uncertainty estimation in DNNs [4, 5]. They are used as prior distributions over some target function $f$, *i.e.*, $f \sim \mathcal{GP}(\mu(\cdot), K_\phi(\cdot, \cdot))$, where $\mu(\cdot)$ and $K_\phi(\cdot, \cdot)$ are, respectively, the mean and covariance functions of the GP. Deep Gaussian Processes (DGPs) [6] concatenate several GPs across a layered network, defining a hierarchical structure that leads to a more flexible probabilistic model. This concatenation enlarges the class of functions that can be modeled, allowing to capture complex patterns within the data.

Bayesian inference in DGPs is intractable. Thus, approximations using variational inducing points such as Double Stochastic Variational Inference (DSVI) [7] must be employed. To alleviate optimization difficulties, two main tools are often used: (i) a whitened representation of the process values at the inducing points [8] to enhance the numerical stability of the model [9–11], and (ii) an identity mean function in the inner layers of the model [7]. In [7], it is claimed that the identity mean function is needed to avoid the pathologies outlined in [12]. However, the reality is that, in practice, this mean function is used even in 2-layer DGPs, which are far from being similar models to those in [12].

Here, we show, via experiments in a toy dataset and a dataset from the UCI repository, that the need for the identity mean function in DGPs is linked to the initial variational parameters and the usage of the whitened representation. Specifically, we found that a zero mean DGP, with the variational initialization used in [7, 9–11], can lead to mode collapse during the optimization process, which is undesirable. Namely, the algorithm sets the variational mean and covariance to those of the GP prior (a standard Gaussian in the whitened case). Then, a huge noise variance is set in the observation model to explain the observed data, which is considered pure noise. This is not a desirable behavior.

Our main contributions are: 1) we explain why the whitened representation, beyond being designed to enhance mixing in MCMC algorithms [8], provides numerical stability in the Variational's Inference optimization process; 2) we highlight why the identity mean function in DGPs avoids the mode collapse effect that occurs when the zero mean function is used; 3) we provide a theoretical explanation of the effect of whitening in the optimization process; and, 4) we propose a new initialization of the variational parameters of the zero mean DGP that alleviates the mode collapse problem.

## 2 Mode collapse in variational DGPs

Consider the observed data $\mathcal{D} = \{(\mathbf{x}_i, y_i) \mid \mathbf{x}_i \in \mathbb{R}^D, y_i \in \mathbb{R}, i = 1, \dots, N\}$ which is grouped into a matrix $\mathbf{X} = (\mathbf{x}_1, \cdots, \mathbf{x}_N)$ and a vector $\mathbf{y} = (y_1, \cdots, y_N)^\mathrm{T}$. Consider a single GP. The process values at the training points are denoted by $\mathbf{f} = (f(\mathbf{x}_1), \cdots, f(\mathbf{x}_N))^\mathrm{T}$. The set of $M$

Workshop on Bayesian Decision-making and Uncertainty, 38th Conference on Neural Information Processing Systems (NeurIPS 2024).

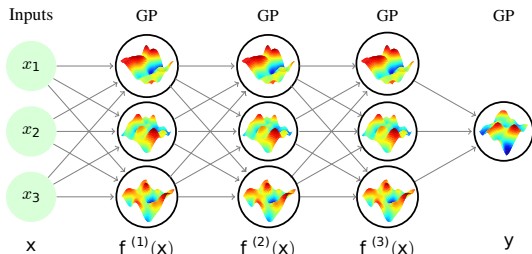

Figure 1: Example of a three layer DGP, with three units per layer.

inducing locations $\mathbf{Z} = (\mathbf{z}_1, \cdots, \mathbf{z}_M)$, with $\mathbf{z}_i \in \mathbb{R}^D$, have their corresponding inducing values $\mathbf{u} = f(\mathbf{Z}) = (\mathbf{u}^1, \cdots, \mathbf{u}^M)^\mathsf{T}$. The posterior distribution is approximated using a variational distribution $q(\mathbf{f}, \mathbf{u}) = p(\mathbf{f} \mid \mathbf{u})q(\mathbf{u})$, with $q(\mathbf{u}) = \mathcal{N}(\mathbf{u} \mid \mathbf{m}, \mathbf{S})$ where $\mathbf{m}$ and $\mathbf{S}$ are the *variational parameters* [13]. The prior over $\mathbf{u}$ is $\mathcal{N}(\mu_\mathbf{Z}, K_\mathbf{ZZ})$. In the whitened representation, we rewrite $\mathbf{u} = \mathbf{L}_\mathbf{ZZ}\mathbf{v} + \mu_\mathbf{Z}$ where $K_\mathbf{ZZ} = \mathbf{L}_\mathbf{ZZ}\mathbf{L}_\mathbf{ZZ}^T$ and $p(\mathbf{v}) = \mathcal{N}(\mathbf{0}, \mathbf{I})$. Then, we make inference about $\mathbf{v}$ instead of $\mathbf{u}$ and learn $q(\mathbf{v}) = \mathcal{N}(\mathbf{m}_\mathbf{v}, \mathbf{S}_\mathbf{v})$. Variational parameters are initialized as $\mathbf{m} = \mathbf{m}_\mathbf{v} = 0, \mathbf{S} = \mathbf{S}_\mathbf{v} = 10^{-5}\mathbf{I}$.

DGPs concatenate layers of GPs as observed in Fig. 1. Adding the superscript $l$ to denote the $l$-th layer, the training objective for the DSVI algorithm, described in [7], is the ELBO:

$$\mathcal{L} = \sum_{n=1}^N \mathbb{E}_{q(\mathbf{f}_n^L)} \left[\log p(y_n \mid \mathbf{f}_n^L)\right] - \sum_{l=1}^L \mathrm{KL}\left(q(\mathbf{u}^l) \parallel p(\mathbf{u}^l; \mathbf{Z}^l)\right) . \tag{1}$$

**Non-whitened GPs:** In classic SVGPs, the mean of the marginal variational distribution $q(\mathbf{f}^{l=1})$ is[1]:

$$\mu_{qf} = K_\mathbf{XZ}K_\mathbf{ZZ}^{-1}\mathbf{m} + \mu_\mathbf{X} - K_\mathbf{XZ}K_\mathbf{ZZ}^{-1}\mu_\mathbf{Z} . \tag{2}$$

The KL between prior and posterior is given by:

$$\mathrm{KL}\left(q(\mathbf{u}) \parallel p(\mathbf{u})\right) = \frac{1}{2}\left[\log(|K_\mathbf{ZZ}||\mathbf{S}|^{-1}) - M + \mathrm{Tr}\left(K_\mathbf{ZZ}^{-1}(\mathbf{S} + (\mathbf{m} - \mu_\mathbf{Z})(\mathbf{m} - \mu_\mathbf{Z})^T)\right)\right] . \tag{3}$$

**Whitened GPs:** In the whitened representation we have:

$$\mu_{qf}^w = K_\mathbf{XZ}[L_\mathbf{ZZ}^T]^{-1}\mathbf{m}_\mathbf{v} + \mu_\mathbf{X} , \tag{4}$$

$$\mathrm{KL}\left(q(\mathbf{v}) \parallel p(\mathbf{v})\right) = \frac{1}{2}\left[-\log|\mathbf{S}_\mathbf{v}| - M + \mathbf{m}_\mathbf{v}^T\mathbf{m}_\mathbf{v} + \mathrm{Tr}(\mathbf{S}_\mathbf{v})\right] . \tag{5}$$

**KL at initialization and mode collapse:** In the whitened case, at initialization, the marginal distribution's variational mean for a zero mean DGP is zero and the KL is very close to zero . Thus, the minimization of the objective will likely force $\mathbf{m}_\mathbf{v} = 0$ and $\mathbf{S}_\mathbf{v} = \mathbf{I}$ (since it is the KL minimizer), and will minimize the ELL, *i.e.*, the data-dependent term in (1), by learning a huge observation noise. However, when using an identity mean function it is simpler to optimize the ELL, since the posterior mean at initialization is $\mathbf{x}_i$, the optimization forces $\mathbf{m}_\mathbf{v}$ to move away from its initial value 0 to learn a map from $\mathbf{x}_i$ to the outputs $\mathbf{y}$. This often avoids mode collapse. For the non-whitened case, at initialization, the variational mean is also 0 in the zero mean DGP. However, we should expect the model not to be so prone to mode collapse, since at initialization the KL depends on $K_\mathbf{ZZ}$ via the prior, which can differ from the identity matrix (depending on the length-scale value). The contrary may also be true, *i.e.*, one may also expect mode collapse, since now we can freely adapt both the non-whitened GP prior kernel hyper-parameters and $\mathbf{Z}$, and the variational parameters $\mathbf{m}$ and $\mathbf{S}$ to minimize KL. Therefore, in the non-whitened case, mode collapse depends not only on the initialization of $q$, but on the initial kernel's hyper-parameter as well. In any case, this model usually suffers from optimization difficulties, so even though mode collapse can be controlled, the model is not useful in practice as it is complicated to optimize.

**Optimization difficulties:** There is a different impact on the KL divergence depending on whether the whitened representation is used or not. In the whitened representation one often observes that: 1) the KL at initialization is close to zero, providing a small learning signal to the objective; and

---

[1]When $l > 2$, $\mathbf{X}$ is replaced by points $\mathbf{f}^{l-1}$ sampled from $q(\mathbf{f}^{l-1})$ at the previous layer.

2) the KL only depends on the variational parameters $\mathbf{m}$ and $\mathbf{S}$. By contrast, in the non-whitened representation, the KL divergence at initialization is far from zero, both for the zero mean and even more for the identity mean function (due to the term $\mathbf{m} - \mu_{\mathbf{Z}}$). Furthermore, the KL depends on both the variational and the model parameters, *e.g.*, $\mathbf{Z}$, length-scales, etc. These two differences complicate optimization and we hypothesize they are the reason why one observes oscillations in the objective during optimization, making the whitened representation the standard one used both in variational GPs and DGPs [9–11], see Fig. 6 in the Appendix. [2]

## 2.1 Variational DGPs in practice

Understanding the implementation of DGPs is key to a comprehensive view of their optimization process. In GPFLOW, the library used in our experiments, the whitened representation is implemented as outlined above. However, the non-whitened representation is implemented using the reparameterization $\mathbf{m}' = \mathbf{m} - \mu_{\mathbf{Z}}$, which changes the predictive mean and KL divergence to:

$$
\mu_{qf} = K_{\mathbf{XZ}} K_{\mathbf{ZZ}}^{-1} \mathbf{m}' + \mu_{\mathbf{X}} \,,
$$
$$
\mathrm{KL}\left(q(\mathbf{u}) \parallel p(\mathbf{u})\right) = \frac{1}{2}\left[ -\log|\mathbf{S}| - M + \log|K_{\mathbf{ZZ}}| + \mathrm{Tr}\left( K_{\mathbf{ZZ}}^{-1}(\mathbf{S} + \mathbf{m}'\mathbf{m}'^{T}) \right) \right]. \tag{6}
$$

This reparameterization simplifies the implementation since both whitened and non-whitened models require KL computation between q and a zero mean prior. Nevertheless, the optimization difficulties outlined above persist due to the term $K_{\mathbf{ZZ}}$ in the KL. For the rest of the work, unless explicitly mentioned, we will be using the aforementioned reparameterized version of the variational mean.

## 3 Avoiding mode collapse via data-driven initialization

We now present a novel initialization that mimics the effect of the identity mean function in DGPs but uses a zero mean function. We achieve this by selecting appropriate values for $\mathbf{m}$, $\mathbf{m}_{\mathbf{v}}$ and $\mathbf{Z}$.

**Inner layers:** At initialization, in the inner layers ($l = 1, \cdots, L-1$), when using the identity mean function, the mean of the posterior distribution takes the value $\mu_{qf}^{w} = \mu_{qf} = \mathbf{x}$ for any point $\mathbf{x} \in \mathbf{X}$. We aim at selecting $\mathbf{m}$ and $\mathbf{m}_{\mathbf{v}}$ (in the whitening case) such that, at initialization, we obtain the same posterior mean *while* using a zero mean DGP. Assuming a 1-dimensional input space (although this can be generalized to arbitrary dimensions), this is achieved by solving the linear systems:

$$
\mathbf{X} = K_{\mathbf{XZ}} K_{\mathbf{ZZ}}^{-1} \mathbf{m} + \underbrace{\mu_{\mathbf{X}} - K_{\mathbf{XZ}} K_{\mathbf{ZZ}}^{-1} \mu_{\mathbf{Z}}}_{0} \,, \quad \text{and} \quad \mathbf{X} = K_{\mathbf{XZ}}[L_{\mathbf{ZZ}}^{T}]^{-1} \mathbf{m}_{\mathbf{v}} + \underbrace{\mu_{\mathbf{X}}}_{0} \tag{7}
$$

These are linear equations in $\mathbf{m}$ and $\mathbf{m}_{\mathbf{v}}$, that depend on three independent variables ($\mathbf{m}, \mathbf{X}, \mathbf{Z}$.). To solve them, $\mathbf{X}, \mathbf{Z}$ must be given some value, requiring $\dim(\mathbf{X}) = \dim(\mathbf{Z})$ to yield a solvable system. We fix the inducing locations $\mathbf{Z}$ to a subset of the train data, *i.e.*, $\mathbf{Z} \subset \mathbf{X}$. This choice gives the solutions $\mathbf{m} = \mathbf{Z}$, and $\mathbf{m}_{\mathbf{v}} = \mathbf{L}_{\mathbf{ZZ}}^{-1} \mathbf{Z}$, and it guarantees that the initialization will predict identity values at the inducing points, *i.e.*, when $\mathbf{X} = \mathbf{Z}$. In both cases, we initialize $\mathbf{S} = 10^{-5}\mathbf{I}$ so that the initial forwarded samples in DSVI do not differ from the mean.

**Output layer:** DGPs usually use a zero mean GP in its output layer with $\mathbf{m}, \mathbf{m}_{\mathbf{v}}$ initialized to $\mathbf{0}$. We propose to extend the idea of the inner layers to the output layer. However, in this case, we want the posterior mean to resemble the target $\mathbf{y}$ values[3]. To this end, we replace $\mathbf{X}$ by $\mathbf{y}$ in both l.h.s. of Eq. (7). This gives $\mathbf{m} = \mathbf{y}$, and $\mathbf{m}_{\mathbf{v}} = \mathbf{L}_{\mathbf{ZZ}}^{-1}\mathbf{y}$, when using whitening. By selecting the inducing locations $\mathbf{Z}$ as a subset of $\mathbf{X}$, we obtain precise posterior predictive means at those data points at initialization. This leads to an initial good solution when using enough inducing points, see Fig. 2.

Let us remark the difference between our proposed initialization and that of standard DGP configurations. Consider a non-whitened DGP with the aforementioned reparameterization of the variational mean $\mathbf{m}'$. Assume an identity mean function in the inner layers, a zero mean function in the output

---

[2]We have not explicitly found that these are the reasons why these libraries use whitening in practice, beyond being recommended in its source code, as seen for example in [11] in *this comment*.

[3]In classification we would use a one-hot encoding of the class labels.

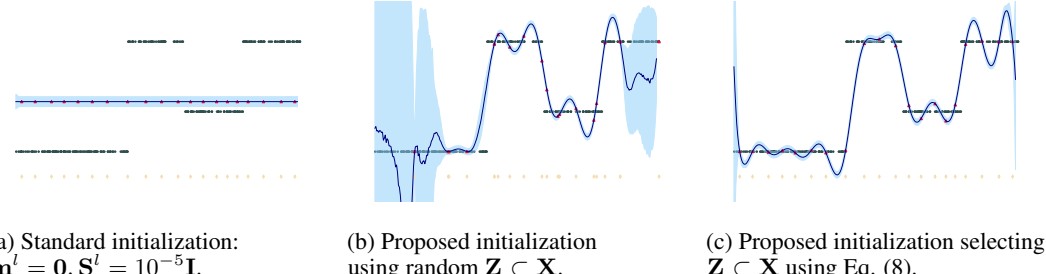

(a) Standard initialization: $\mathbf{m}^l = \mathbf{0}, \mathbf{S}^l = 10^{-5}\mathbf{I}$.

(b) Proposed initialization using random $\mathbf{Z} \subset \mathbf{X}$.

(c) Proposed initialization selecting $\mathbf{Z} \subset \mathbf{X}$ using Eq. (8).

Figure 2: Initialization predictions of whitened 2 layer DGPs. Yellow dots indicate the inducing locations $\mathbf{Z}$ and red dots indicate predictive mean $\mathbf{m_v}$ at the inducing locations.

layer, and that we initialize $\mathbf{m}' = \mathbf{0}$. This is the typical setup considered in the literature for DGPs [7]. The output of the inner layers of this model model at $\mathbf{Z}$, is $\mathbf{Z}$, *i.e.*, the identity, as specified by the prior mean $\mu_{\mathbf{Z}}$. See Eq. (6). However, since the identity mean function is not used in the last layer [7], the predictive mean at the inducing points $\mathbf{Z}$ will be zero, unlike in the proposed method, described in the previous paragraph. Something similar will happen in the case of the whitened representation, when $\mathbf{m_v} = \mathbf{0}$. See Eq. (4). That is, the initial DGP predictive mean, without training the model, will be zero for every input.

**Inducing points:** The initial inducing points may impact the initial solution. Selecting them at random from the training set may lead to regions of the input space being unrepresented, leading to a poor initial solution in those areas. See Fig. 2. To ensure a good initialization, we use a two-step algorithm:

1. We compute $M$ centroids $\mathcal{C} = \{\mathbf{c}_1, \cdots, \mathbf{c}_M\}$ of $\mathbf{X}_{\text{train}}$ using *k-means*.
2. We select the inducing locations $\mathbf{z}_j$ from $\mathbf{X}_{\text{train}}$ using the cosine distance:

$$\mathbf{z}_j = \arg\min_{\mathbf{x}_j \in \mathbf{X}_{\text{train}}} d(\mathbf{c}_j, \mathbf{x}_j) = \arg\min_{\mathbf{x}_j \in \mathbf{X}_{\text{train}}} \mathbf{c}_j \cdot \mathbf{x}_j / (\|\mathbf{c}_j\| \|\mathbf{x}_j\|), \quad \forall \mathbf{c}_j \in \mathcal{C}. \tag{8}$$

Fig. 2 compares the predictive distribution (without training) of the standard initialization of a DGP with that of the proposed initialization with a random selection of the inducing points among the training points, and when selecting the inducing locations as described above. We observe that the proposed initialization already explains quite well the observed data with no training of the objective.

**Optimization difficulties:** As mentioned, with the standard initialization, the KL in the whitened case provides a more stable learning signal (Sec. 2). In the proposed initialization, in the whitened case we have $p(\mathbf{v}) = \mathcal{N}(0, \mathbf{I})$ and $q(\mathbf{v}) = \mathcal{N}(\mathbf{L}_{\mathbf{ZZ}}^{-1}\mathbf{Z}, 10^{-5} \cdot \mathbf{I})$. In the non-whitened (reparameterized version) we have $p(\mathbf{u}) = \mathcal{N}(0, K_{\mathbf{ZZ}})$ and $q(\mathbf{u}) = \mathcal{N}(\mathbf{Z}, 10^{-5} \cdot \mathbf{I})$. Therefore, the whitened parameterization is expected to lead to a more stable optimization process since, although both KL are different from 0 at initialization, the gradient of the KL for the non-whitened case depends on both the model parameters (*e.g.*, length-scales and inducing points) and the variational parameters.

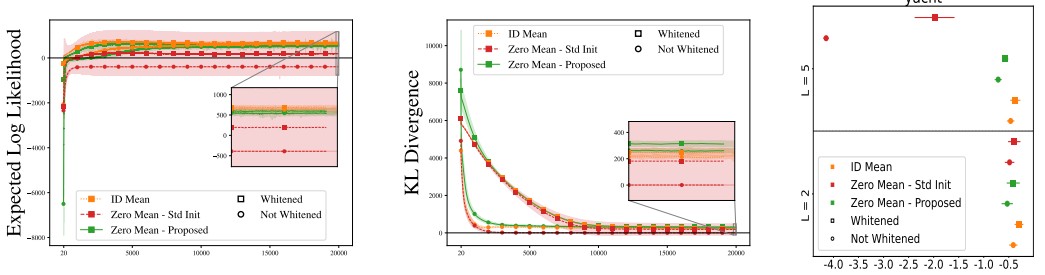

Figure 3: **Left and middle**: ELL and KL, respectively, during train of the 5 layer, zero mean function DGP in the *Yacht* dataset. The mean and standard deviation across 20 splits are plotted. **Right**: Log likelihood results (right is better) of the $L = 2$ and $L = 5$ layer DGPs in *Yacht*. The model with 5 layers and zero mean suffers from mode collapse, which is solved by the proposed initialization.

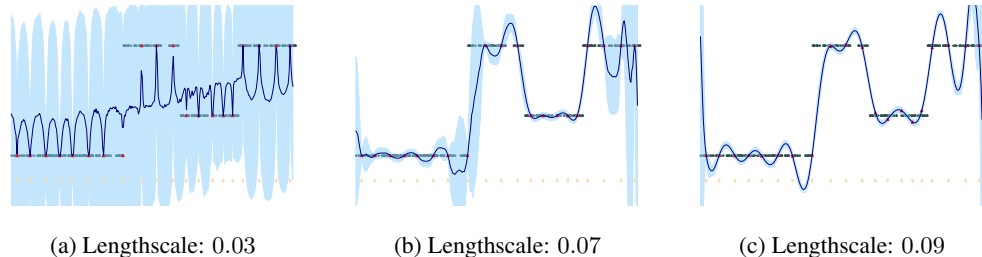

(a) Lengthscale: 0.03        (b) Lengthscale: 0.07        (c) Lengthscale: 0.09

Figure 4: Predictive distribution of the proposed initialization when the length-scale varies. Proper kernel initialization is key for the success of this initialization.

## 4    Experiments

To validate the proposed initialization, we compare it with the standard initialization. We perform an extensive qualitative evaluation on the toy dataset presented in Fig. 2 and on the *Yatch* dataset. The results show that the proposed initialization has two benefits: a) it leads to a faster convergence than the standard initialized DGP; and b) the proposed initialization avoids mode collapse when using a 5-layer DGP. Fig. 5 in Appendix B.1 supports point a). The next paragraph supports point b).

We perform an extensive analysis of 8 different UCI datasets. See Appendix B.2. The most interesting case is the *Yacht* dataset analyzed below. The data-dependent term of the objective (ELL), as a function of the training epochs, is shown in Fig. 3 (left). The KL is also reported (center). We observe a clear example of the mode collapse in the 5-layer DGP when using the zero mean function with the standard initialization. Specifically, the test log-likelihood results displayed in Fig. 3 (right) are significantly worse for the zero mean function with the standard initialization DGPs. We observe the ELL and the KL (left and center of the figure). In the standard initialization, when using the zero mean function, the KL quickly falls to zero when whitening is not used. Besides this, we also observe that the KL has a high variance when the whitened representation is used. This indicates that there are some data splits where mode collapse happens, even when using whitening. By contrast, the proposed initialization solves the mode collapse problem, preventing the KL from falling to zero by choosing a good initial solution. It also leads to an overall higher final KL. The ELL chart (left) shows that, in the whitened case, our proposal exhibits a fast convergence and the final results pair up with the DGP with identity mean function. The non-whitened case has a slower convergence because the initial solution has a low ELL, caused by the initial value of the kernel's length-scale[4]. We observe this behavior in other datasets. Appendix B.2 has a figure with all the results of LL and RMSE.

### 4.1    Kernel invertibility dependency

As a limitation of the proposed initialization, we remark that the correct inversion of the kernel evaluated at the inducing points highly influences the initial solution. The parameters of the kernel (in our case, the length-scale $\ell$ of the RBF kernel) must be properly selected so that the covariance matrix is not ill-conditioned, making the proposed initialization properly predict the corresponding inducing value at the selected inducing locations. An example of this can be observed in Fig. 4. However, this limitation can be surpassed by performing an initial evaluation of the model using different values of the kernel length-scale and selecting the one with higher ELL or lower RMSE.

## 5    Conclusions

DGPs may suffer from mode collapse (*i.e.*, convergence to the prior) as the number of layers grows. Here, we have presented an initialization of SVGPs and DGPs that uses the training data. This initialization fixes the inducing points and the variational parameters to predict an initial good solution of the targets. The results obtained, when using this initialization, show that it has benefits both in the convergence speed DGPs and in avoiding the mode collapse problem. A limitation is that it requires solving a linear system that depends on the initial kernel parameters, which may be problem-dependent.

---

[4]In the UCI datasets we have not selected a good initial kernel length-scale for each dataset. Due to this, the faster convergence is not appreciated in Fig. 3.

## Acknowledgments and Disclosure of Funding

The authors acknowledge financial support from project PID2022-139856NB-I00, funded by MCIN and from the Autonomous Community of Madrid (ELLIS Unit Madrid). They also acknowledge the use of the facilities of Centro de Computación Científica, UAM. This work was also supported by grant PID2022-140189OB-C22 funded by MCIN/AEI/10.13039/501100011033 and by "ERDF A way of making Europe", by the European Union.

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

# A  Background

We include an extended background section for a clearer understanding of the problem statement and notation.

We tackle the standard regression problem, where the goal is to infer an unknown function $f : \mathbb{R}^D \to \mathbb{R}$, using the observed data $\mathcal{D} = \{(\mathbf{x}_i, y_i) \mid \mathbf{x}_i \in \mathbb{R}^D, \ y_i \in \mathbb{R}, \ i = 1, \ldots, N\}$. We denote $\mathbf{X} = (\mathbf{x}_1, \cdots, \mathbf{x}_N)$ and $\mathbf{y} = (y_1, \cdots, y_N)$. Gaussian Processes [1] place a multivariate Gaussian prior distribution over $f$, that is, $\mathbf{f} = (f(\mathbf{x}_1), \cdots, f(\mathbf{x}_N)) \sim \mathcal{N}(\mu(\mathbf{X}), K_\phi(\mathbf{X}, \mathbf{X}))$, where $\mu : \mathbb{R}^D \to \mathbb{R}$ is the mean function, and $K_\phi(\mathbf{X}, \mathbf{X})$ is the covariance matrix determined by a covariance function $K_\phi : \mathbb{R}^D \times \mathbb{R}^D \to \mathbb{R}$. We write $\mu_{\mathbf{x}} = \mu(\mathbf{x})$ and $K_{\mathbf{xx}} = K_\phi(\mathbf{x}, \mathbf{x})$ to shorten notation. The exact posterior distribution $p(\mathbf{f} \mid \mathcal{D})$ is intractable since it scales cubically with $N$, so we use sparse variational Gaussian Processes (SVGPs) [13] to scale to large datasets.

SVGPs use a set of inducing locations $\mathbf{Z} = (\mathbf{z}_1, \cdots, \mathbf{z}_M)$, with $\mathbf{z}_i \in \mathbb{R}^D$, and $M \ll N$ and their corresponding inducing values $\mathbf{u} = f(\mathbf{Z}) = (\mathbf{u}^1, \cdots, \mathbf{u}^M)$, where $p(\mathbf{u}) = \mathcal{N}(\mathbf{u} \mid \mu(\mathbf{Z}), K_\phi(\mathbf{Z}, \mathbf{Z}))$. The posterior distribution is approximated using a variational distribution $q(\mathbf{f}, \mathbf{u}) = p(\mathbf{f} \mid \mathbf{u})q(\mathbf{u})$, where $q(\mathbf{u}) = \mathcal{N}(\mathbf{u} \mid \mathbf{m}, \mathbf{S})$ where $\mathbf{m}$ and $\mathbf{S}$ are the *variational parameters*. Using this variational distribution, the posterior distribution is Gaussian $q(\mathbf{f}) = \mathcal{N}(\mathbf{f} \mid \mu_{qf}, K_{qf})$, where:

$$\mu_{qf} = K_{\mathbf{XZ}} K_{\mathbf{ZZ}}^{-1} \mathbf{m} + \mu_{\mathbf{X}} - K_{\mathbf{XZ}} K_{\mathbf{ZZ}}^{-1} \mu_{\mathbf{Z}} \tag{9}$$

$$K_{qf} = K_{\mathbf{XX}} - K_{\mathbf{XZ}} K_{\mathbf{ZZ}}^{-1} K_{\mathbf{ZX}} + [K_{\mathbf{XZ}} K_{\mathbf{ZZ}}^{-1}] \mathbf{S} [K_{\mathbf{XZ}} K_{\mathbf{ZZ}}^{-1}]^T \tag{10}$$

The presented formulation is sometimes referred to as the *unwhitened* representation of a GP. Another common form is the *whitened prior* GP representation [8, 14]. This consists of expressing $\mathbf{u} = \mathbf{L}_{\mathbf{ZZ}} \mathbf{v}$ with $\mathbf{L}_{\mathbf{ZZ}} \mathbf{L}_{\mathbf{ZZ}}^T = K_{\mathbf{ZZ}}$ and $\mathbf{v} = \mathcal{N}(\mathbf{v} \mid \mathbf{0}, \mathbf{I})$. The original variational parameters are recovered as $\mathbf{m} = \mu_{\mathbf{Z}} + \mathbf{L}_{\mathbf{ZZ}} \mathbf{m}_{\mathbf{v}}$ and $\mathbf{S} = \mathbf{L}_{\mathbf{ZZ}} \mathbf{S}_{\mathbf{v}} \mathbf{L}_{\mathbf{ZZ}}^T$, so the parameters to estimate are now $\mathbf{m}_{\mathbf{v}}, \mathbf{S}_{\mathbf{v}}$. Using this representation, the posterior $q^w(\mathbf{f}) = \mathcal{N}(\mathbf{f} \mid \mu_{qf}^w, K_{qf}^w)$ is computed as:

$$\mu_{qf}^w = K_{\mathbf{XZ}} [\mathbf{L}_{\mathbf{ZZ}}^T]^{-1} \mathbf{m}_{\mathbf{v}} + \mu_{\mathbf{X}}, \tag{11}$$

$$K_{qf}^w = K_{\mathbf{XX}} - K_{\mathbf{XX}} K_{\mathbf{ZZ}}^{-1} K_{\mathbf{ZX}} + K_{\mathbf{XZ}} \mathbf{L}_{\mathbf{ZZ}}^{T\ -1} \mathbf{S}_{\mathbf{v}} \mathbf{L}^{-1} K_{\mathbf{ZX}}. \tag{12}$$

Whitening the GP prior leads to a much faster convergence in many cases [14].

**Deep Gaussian Processes**

Following the fashion of deep neural networks, DGPs concatenate layers of GPs to generalize them and extend their flexibility. Formally, the output of the GP in layer $l - 1$ is used as the input of layer $l$, defining a prior recursively on each layer $F^1, \cdots, F^L$. Each layer $l$ is defined by its inducing locations and values $(\mathbf{Z}^l, \mathbf{u}^l)$, and the corresponding variational parameters $(\mathbf{m}^l, \mathbf{S}^l)$. The joint distribution of the DGP factorizes as:

$$p(\mathbf{y} \mid \{\mathbf{F}^l, \mathbf{u}^l\}_{l=1}^L) = \underbrace{\prod_{i=1}^N p(y_i \mid \mathbf{f}_i^L)}_{\text{likelihood}} \underbrace{\prod_{l=1}^L p(\mathbf{F}^l \mid \mathbf{u}^l; \ \mathbf{F}^{l-1}, \mathbf{Z}^{l-1}) p(\mathbf{u}^l; \mathbf{Z}^{l-1})}_{\text{DGP Prior}}. \tag{13}$$

The exact posterior remains intractable in DGPs, so it is approximated using a variational distribution that mimics the one in SVGPs and assumes layer independence:

$$q(\{\mathbf{F}^l, \mathbf{u}^l\}_{l=1}^L) = \prod_{l=1}^L p(\mathbf{F}^l \mid \mathbf{u}^l; \mathbf{F}^{l-1}, \mathbf{Z}^{l-1}) q(\mathbf{u}^l). \tag{14}$$

With these considerations, we obtain an ELBO that can be optimized using two sources of stochasticity [7]:

$$\mathcal{L} = \sum_{n=1}^N \mathbb{E}_{q(\mathbf{f}_n^L)} \left[ \log p(y_n \mid \mathbf{f}_n^L) \right] - \sum_{l=1}^L \mathrm{KL} \left( q(\mathbf{u}^l) \parallel p(\mathbf{u}^l; \mathbf{Z}^{l-1}) \right) \tag{15}$$

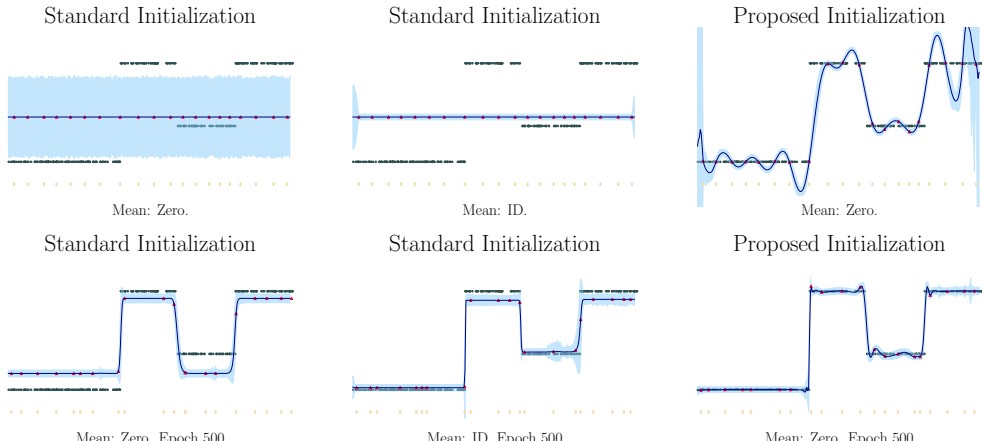

Figure 5: Top row: Initialization of the DGP. Standard initialization for the Zero mean and identity mean, and proposed initialization at the right. Bottom row: predictions of each of the models after 500 epochs.

## B Further Experiments

**Implementation details**  In the toy experiment, we generate 300 datapoints train the models using a learning rate of $\lambda = 10^{-3}$. For the UCI experiments, we use 20 different train-test splits. We use a learning rate of $\lambda = 10^{-2}$ and train for $20K$ iterations. In both cases, we run all our experiments in CPU. For the implementation, we use GPFLOW *2.1.3*, which builds upon *Tensorflow*. The used kernel is the standard RBF kernel:

$$K(\mathbf{x}, \mathbf{x}') = \exp\left(-\frac{\|\mathbf{x} - \mathbf{x}'\|^2}{2\ell^2}\right), \tag{16}$$

where $\ell$ is the kernel *lengthscale*.

### B.1 Toy Dataset

During the experimentation in the toy dataset, we have found other significant results that further show the benefits or limitations of using the proposed initialization. We present them in this section.

**Convergence speed**  Due to the initial good solution, we found out that our initialization speeds up the convergence of DGPs. This comes from the fact that, when the kernel parameters are properly chosen, the initial predictions pose an already good solution, with a lower RMSE. An example of this is shown in Fig. 5.

**Whitening provides stability**  We have observed that for any initialization or mean function, using the whitening representation of the DGP is key to optimization smoothness. Even in the case of the Identity mean function, where the convergence is always faster than the Zero mean DGP, we have observed in this toy dataset that not using the whitening representation may lead to a poor training of the model. An example of this is shown in Fig. 6, where the KL of the 5 layer, Identity mean DGP presents oscillations that hinder the optimization task and do not appear in the whitened case.

### B.2 UCI Datasets

We compare our initialization with the standard one, using both whitened and non-whitened representations of the variational distribution. We also include the Identity mean DGP for further comparison. We use $L \in \{2, 3, 4, 5\}$ for each model.

The Log Likelihood results are shown in Fig. 7. The results show that DGPs with standard initialization suffers from mode collapse when the number of layers is increased (which can be seen in *Kin8nm, Power, Redwine or Yatch*). However, the proposed initialization prevents that mode collapse, obtaining

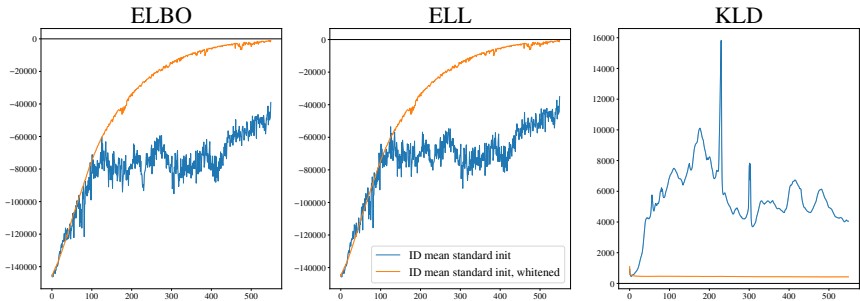

Figure 6: Loss function values of the $5-$ layer DGP in the toy dataset. The convergence stability that the whitened version of the model provides is clearly appreciated. Both the ELL and KL are much harder to optimize in the non-whitened case.

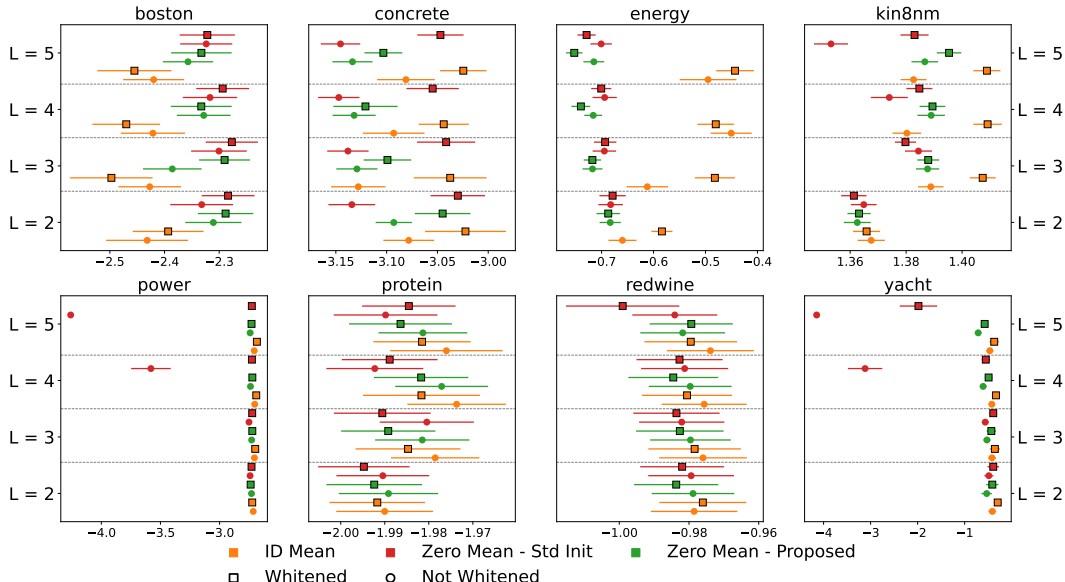

Figure 7: Log likelihood (right is better) results in the 8 UCI dataset, for 4 different number of layers.

results which are at least paired to the results obtained using the identity mean function in the inner layers of the DGP.

In Fig. 8 we represent the results using the RMSE. The conclusions are similar to the ones obtained from the LL results. Again, the poor solutions in terms of RMSE can be clearly seen in the case of a zero mean DGP with standard initialization, with those cases being solved with our initialization.

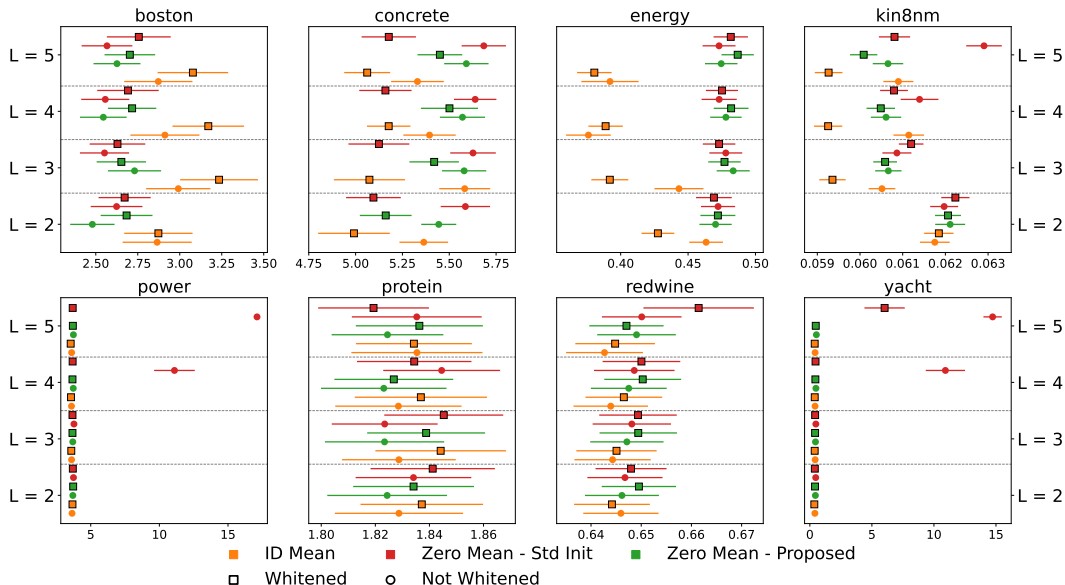

Figure 8: RMSE (left is better) results in the 8 UCI dataset, for 4 different number of layers.

