# OpenReview forum: "Mode Collapse in Variational Deep Gaussian Processes"
_NeurIPS.cc/2024/Workshop/BDU — NeurIPS BDU Workshop 2024 Poster_

### Official Review · Reviewer_BLVo · 2024-09-16
**Overall is a good paper to be accept for workshop.**

**Rating:** 6
**Confidence:** 4

**Review:**

The document discussing the issue of mode collapse in Variational Deep Gaussian Processes (DGPs) and proposing a solution to this problem. An evaluation based on the given context is as follows:

Quality:

The addressed problem is well identified within a specific area of interest, Bayesian machine learning. In particular, the problem related to mode collapse for DGPs. It clearly presents the problem statement and the proposed solution to that effect, which is a new initialization strategy for the variational parameters. The paper presents experimental results to substantiate its claims; typically, such work implies good quality research.

Clearness:

Appearantly, the paper is well-structured: it introduces the problem, it elaborates on the proposed solution, and discusses the findings in a conclusive manner. The use of figures and tables (not included in the text provided) suggests that the authors have made effort to present their results in accessible ways.

Originality:

The paper claims to develop a new initialization strategy that prevents mode collapse, which would be an original contribution to the field. Another contribution is the theoretical explanation of the whitening effect in the optimization process, something rather new.

Impact:

The work can be impactful since it will address one known problem with DGPs, which are powerful models for function approximation and uncertainty estimation. In case the proposed solution works well, we will have more stable and reliable DGP models, adding value to the machine learning community.

prors:
- The paper tackles a relevant problem in the field of DGPs and provides a potential solution.
- It includes experimental validation, which adds credibility to the proposed method.
- The paper appears to be well-written and structured, making it accessible to readers.

 cons:
- The paper may require a good understanding of Gaussian Processes and variational inference, which could limit its accessibility to a broader audience.
While this is an effective solution, the dependence on the choice of the initial kernel parameters could be a flaw if not handled appropriately.
 Due to the lack of a complete experimental setup and results within the paper, complete robustness testing of the proposed method for different datasets and scenarios cannot be done.

In general, this paper appears to be a good contribution in the Bayesian machine learning area and might enable DGP models to become more stable and give better performance. However, without the complete paper showing details and results of experiments, it would be more appropriate to assess the full significance and impact of the work.

---

### Official Review · Reviewer_qsk6 · 2024-09-28

**Rating:** 6
**Confidence:** 3

**Review:**

The paper investigates mode collapse in Deep Gaussian Processes (DGPs) and proposes a new initialization method to prevent this problem. It identifies the limitations of the current approaches, particularly with the use of identity mean functions, and provides empirical evidence on toy datasets and UCI benchmarks.

**Strengths:**
- Clear Goal: The paper has a well-defined objective, focusing on addressing mode collapse in DGPs and demonstrating the effectiveness of the proposed initialization method.

- Interesting Observations: The insights into why the identity mean function prevents mode collapse are valuable, as well as the exploration of whitening effects on optimization. These observations are potentially useful for practitioners implementing DGPs in real-world applications.

**Weaknesses:**
- Lack of Theoretical Guidance: While the paper provides empirical results, it lacks a strong theoretical framework to guide the proposed approach. The explanations are mainly heuristic and would benefit from a more rigorous theoretical analysis.

- Experiments Limited to Toy Datasets: Although the paper is empirical, most experiments are conducted on toy datasets and a small number of UCI datasets. For broader applicability, more experiments on real-world datasets with larger scales would strengthen the paper's contributions.

---

### Decision · Program_Chairs · 2024-10-09

Accept (Poster)